# Inoculation with *Azospirillum brasilense* as a Strategy to Reduce Nitrogen Fertilization in Cultivating Purple Maize (*Zea mays* L.) in the Inter-Andean Valleys of Peru

**DOI:** 10.3390/microorganisms12102107

**Published:** 2024-10-21

**Authors:** Tatiana Condori, Susan Alarcón, Lucero Huasasquiche, Cayo García-Blásquez, César Padilla-Castro, José Velásquez, Richard Solórzano

**Affiliations:** 1Estación Experimental Agraria Canaán, Dirección de Supervisión y Monitoreo de las Estaciones Experimentales, Instituto Nacional de Innovación Agraria (INIA), Ayacucho 05002, Peru; tati.condori89@gmail.com (T.C.); sumi222015@gmail.com (S.A.); josevelasquez_m@hotmail.com (J.V.); 2Estación Experimental Agraria Donoso, Dirección de Desarrollo Tecnológico Agrario, Instituto Nacional de Innovación Agraria (INIA), Lima 15200, Peru; lucero.26.lhs@gmail.com; 3Escuela de Agronomía, Facultad de Ciencia Agrarias, Universidad Nacional San Cristóbal de Huamanga (UNSCH), Ayacucho 05001, Peru; cayo.garciablasquez@unsch.edu.pe; 4Estación Experimental Agraria Pucallpa, Dirección de Supervisión y Monitoreo de las Estaciones Experimentales, Instituto Nacional de Innovación Agraria (INIA), Ucayali 25002, Peru; cesar.padillacastro@outlook.com; 5Dirección de Supervisión y Monitoreo en las Estaciones Experimentales Agrarias, Instituto Nacional de Innovación Agraria (INIA), Av. La Molina 1981, Lima 15024, Peru; 6Facultad de Ciencias Ambientales, Universidad Científica del Sur (UCSUR), Lima 15067, Peru

**Keywords:** purple corn, PGPR, microbial inoculation, nitrogen

## Abstract

Purple maize has gained global significance due to its numerous nutraceutical benefits. However, sustaining its production typically requires high doses of nitrogen fertilizers, which, when applied in excess, can contaminate vital resources such as soil and water. Inoculation with nitrogen-fixing microorganisms, such as those from the *Azospirillum* genus, has emerged as an alternative to partially or fully replace nitrogen fertilizers. This study aimed to evaluate the inoculation effect with *A. brasilense* and varying nitrogen fertilization levels on the yield and quality of purple maize. The experiment was carried out using a randomized complete block design (RCBD) with a 2 × 5 factorial arrangement and five replications. Treatments comprised two inoculation levels (control without inoculation and inoculation with *A. brasilense*) under five nitrogen doses (0, 30, 60, 90, and 120 kg∙ha^−1^, applied as urea). Inoculation with *A. brasilense* resulted in a 10.5% increase in plant height, a 16.7% increase in root length, a 21.3% increase in aboveground fresh biomass, a 30.1% increase in root fresh biomass, and a 27.7% increase in leaf nitrogen concentration compared to the non-inoculated control. Regarding yield, the inoculated plants surpassed the control in both purple maize yield (kg∙ha^−1^) and cob weight by 21.8% and 11.6%, respectively. Across all fertilization levels and parameters assessed, the inoculated treatments outperformed the control. Furthermore, for parameters, namely plant height, leaf nitrogen content, and cob dimensions (length, diameter, and weight), the *A. brasilense* inoculation treatment with 90 kg N∙ha^−1^ was statistically equivalent or superior to the non-inoculated control with 120 kg N∙ha^−1^. These results indicate that inoculation with *A. brasilense* positively impacted purple maize at all nitrogen levels tested and improved nitrogen use efficiency, enabling a reduction of 30 kg N∙ha^−1^ without compromising performance in key parameters.

## 1. Introduction

Maize (*Zea mays* L.) is one of the oldest known cereal grains, with a globally distributed production [1]. Purple maize has gained significant global recognition among its pigmented varieties, despite not being consumed as common table maize [2]. Its markets now extend to Asia, the United States, and Europe [3], largely due to its high anthocyanin content. This compound contributes to its applications in the medical field [1,4] and in the textile, cosmetic, and food industries [5].

In Peru, purple maize production and harvested area grew by 2.7% and 2.4%, respectively, between 2016 and 2020, with Lima and Ayacucho accounting for over 50% of market representation [6]. Among them, Ayacucho experienced the highest growth in purple maize production during this period, attributed to effective agronomic practices, increased technical assistance [6], and favorable climatic conditions. However, managing this crop requires substantial nitrogen inputs to meet grain production demands [7], making it one of the few Andean crops consistently fertilized, typically with large amounts of manure (guano) and urea.

These fertilizers are entirely imported at high costs. In 2022, Peru imported 330,287 tons of urea at USD 1198.89 per ton, which represents a 1.47% increase compared to the previous year [8]. Moreover, the excessive and improper use of nitrogen-based inputs depletes soil fertility, adversely affecting the soil microbiota [9] and contributing to environmental pollution, such as groundwater nitrate accumulation [10]. These challenges, coupled with population growth and climate change, have intensified the pursuit of agricultural productivity and sustainability [11]. This has driven the adoption of agricultural technologies aimed at reducing synthetic fertilizer use or enhancing their efficiency while still ensuring sufficient crop production to meet global food demand.

One promising alternative to address this issue is through the use of nitrogen-fixing microorganisms, also known as diazotrophs [12]. These microorganisms can reduce atmospheric nitrogen, converting it into a plant-available form [13]. Nitrogen fixation can occur symbiotically, as seen in the relationship between rhizobia and legumes, or associatively, with the most studied group being microorganisms from the *Azospirillum* genus.

Several studies have demonstrated the effectiveness of *Azospirillum* inoculation in maize cultivation [14,15,16,17]. One of the most recent was by Contreras-Liza et al. [18], who inoculated *Azospirillum* sp. associated with two levels of nitrogen fertilization in the production of Sweet Corn. They found high significance in the grain yield per hectare of the *Azospirillum* treatments in relation to the control. In another study on corn [19], it was observed that the effect of inoculation associated with different sources and levels of nitrogen fertilizer was beneficial in corn regardless of the nitrogen source. *Azospirillum brasilense* was also used in grasses such as wheat [20] with two sources and doses of nitrogen, verifying the great potential to increase the efficiency of nitrogen fertilizers. However, specific inoculation research in purple maize remains limited, and no studies have been reported on this practice in maize cultivation in Peru. In this context, the present research aimed to evaluate the impact of *Azospirillum brasilense* inoculation at various nitrogen fertilization levels in a purple maize field to improve fertilizer efficiency and reduce applied doses while maintaining crop yield.

## 2. Materials and Methods

### 2.1. Experimental Area Characteristics

The study was carried out from September 2023 to March 2024 in a 1610.4 m^2^ plot located at the Canaán Agricultural Experiment Station (EEA) (13°9′53.87′ S 74°12′14.62′ W, 2735 m.a.s.l.) of the National Institute of Agrarian Innovation (INIA), Andrés Avelino Cáceres Dorregaray district, province of Huamanga, Ayacucho Region. During that period, the average temperature was 18 °C, with a maximum of 25 °C and a minimum of 11 °C, and the relative humidity was 83% (Figure 1). The historical averages were calculated based on the information provided by the INIA Canaán Meteorological Station (13°9′ S; 74°13′ W; 2761 m.a.s.l.) of the National Meteorological and Hydrological Service of Peru (SENAMHI), and the Regional Government of Ayacucho (GRA).

### 2.2. Soil Characteristics

Before sowing, a soil sample was collected at a 30 cm depth and analyzed for texture [21], pH [22], electrical conductivity (EC) [23], organic matter (OM) [21], total nitrogen (N) [24], available phosphorus (P) [21], available potassium (K) [25] and exchangeable cation concentration (CEC) [21] at the Soil, Water and Foliar Laboratory—INIA. The observed physicochemical characteristics (Table 1) indicate that, despite showing high phosphorus availability and medium potassium availability, the soil quality is low because it has a medium level of organic matter, a pH close to 8, which limits nutrient availability, and an exchangeable calcium excess, which results in inadequate cation ratios. It is important to point out that the soils at the Canaan Experimental Centre, where the research and seed production were conducted, exhibit low fertility. This is likely due to continuous agricultural campaigns conducted annually, combined with other standard agricultural practices. These repeated cycles of cultivation consistently extract the same nutrients from the soil, leading to nutrient depletion and, consequently, a “weakening” of the soil and a reduction in its overall fertility over time.

### 2.3. Experimental Design

A completely randomized block design with a factorial arrangement (2 × 5) was employed in this study. The first factor was inoculation with *Azospirillum brasilense*, consisting of two levels: without inoculation (control) and with inoculation. The second factor was nitrogen fertilization doses using urea (46% N), with the following five levels: 0, 30, 60, 90, and 120 kg N∙ha^−1^. Table 2 shows the treatments evaluated in this study. Five blocks were included for each treatment, resulting in a total of 50 experimental units. Each experimental unit comprised an area with six furrows, spaced 0.80 m apart, and 5 m in length.

### 2.4. Crop Management

Seeds of the purple maize variety INIA 615 Negro Canaán were used, donated by the National Soil Programme PP089 of the EEA Canaán. Sowing took place on 5 October 2023, with three seeds per stroke, spaced 0.5 m apart. Before sowing, the seeds were disinfected with Vitavax-300 (1000 g/cylinder). The crop was grown under rain-fed conditions, supplemented by gravity irrigation, and manual weeding was performed. During the vegetative stage, *Diabrotica speciosa* and aphids were controlled by applying Cyperklin 25 EC at 200 cc/cylinder, while *Fusarium* spp. (chupadera fungosa) was managed using Vitavax-300 at 1000 g/cylinder. For pest control, *Spodoptera frugiperda* (cogollero), *Heliothis zea* (mazorquero), and *Dalbulus maidis* (puka puncho) were treated with Beta Baytroide at a 10 mL/1000 m^2^ dose. Harvesting was conducted 182 days after sowing (DAS) when the maize cobs had 22–25% moisture content. The cobs were subsequently dried in a closed environment until the corn grains reached 14% moisture.

### 2.5. Fertilization

Fertilization was determined based on soil analysis and the crop’s nutrient requirements, using a combination of island guano (1500 kg∙ha^−1^), diammonium phosphate (120 kg∙ha^−1^), potassium chloride (50 kg∙ha^−1^), and urea (250 kg∙ha^−1^). All treatments received a base fertilization consisting of diammonium phosphate (18-46-0) at 120 kg∙ha^−1^, potassium chloride (0-0-60) at 50 kg∙ha^−1^, and island guano (11-10.5-1.9) at 1500 kg∙ha^−1^. This provided a total nutrient contribution of 186.6 kg∙ha^−1^ nitrogen (N), 212.6 kg∙ha^−1^ phosphorus pentoxide (P_2_O_5_), and 58.2 kg∙ha^−1^ potassium oxide (K_2_O). The study’s main factor was nitrogen fertilization from urea (46% N), applied at different doses and split into two applications. Table 3 provides detailed information on the urea quantities used and the corresponding nitrogen input at each fertilization level.

### 2.6. Inoculation with Azospirillum brasilense

The *A. brasilense* strain was provided by the Universidad Nacional San Cristobal de Huamanga (UNSCH-FOCAM). The strain was isolated from soft maize roots from the village of Ccerayocc, district of Quinua–Ayacucho, and characterized in the Rhizobiology Laboratory of the same university. This strain was previously studied at a greenhouse level on avocado crops [26].

The inoculant was prepared in NFb liquid medium. After 14 days of incubation, quality control was conducted, which included checking the medium’s pH, ensuring the broth’s purity through Gram staining, assessing bacterial concentration, plating in a specific NFb medium, and performing microscopic counting using a Petroff–Hausser chamber. Once the inoculant met the quality standards, both liquid (NFb medium) and solid (peat-based) inoculants were formulated by inoculating 40 mL into 100 g of peat. After 30 days of maturation, quality control was reassessed, confirming a bacterial concentration of 1 × 10^8^ CFU/g in the inoculant.

For the experiment, inoculation was carried out on purple maize seeds on the same day of sowing, mixing peat-based solid inoculant and liquid inoculant at a 150 g and 100 mL rate for 50 kg of seeds, respectively.

### 2.7. Vegetative Parameters

At 132 das, corresponding to the beginning of the flowering stage, the following parameters were evaluated: plant height, root length, and the fresh and dry weights of the aerial and root parts. To conduct these measurements, 3 plants were randomly selected from each experimental unit, excluding those on the border. Plant height was measured from the collar to the base of the panicle, while root length was measured from the collar to the apex of the main root. Fresh weight was recorded using an electronic balance (Type: Explorer™ Pro Precision; Ohaus; Parsippany-Troy Hills, NJ, USA) with a measurement accuracy of 0.001 g. The plant material was then chopped and dried in an oven for 48 h at 65 °C to determine dry weight. Leaf nitrogen content was assessed after flowering by collecting leaf samples from the lower, middle, and upper sections of the 3 plants. These samples were analyzed for foliar nitrogen content at the Soil, Water, and Foliar Laboratory (LABSAF)-INIA Canaán, using the Kjeldahl method. At the time of harvest, the cob height was measured, selecting 10 plants at random and measuring from the base of the plant to the node where the top cob emerged.

### 2.8. Cob Quality

Cobs were harvested from the plants in the two central furrows of each experimental unit. Subsequently, 10 cobs were randomly selected, and the following parameters were measured: cob length, cob diameter, and grains per cob. Cob length was measured from the base to the apex, while cob diameter was taken at the midpoint using a vernier. A proximate and anthocyanin analysis was performed on the cobs when they reached 14% moisture content. The cobs were manually shelled, and a 500 g sample consisting of both corn grains and cob was sent to the laboratory of the Nutritional Research Institute in Lima, Peru, where the following parameters were evaluated: anthocyanins, proteins, ash, and moisture content.

### 2.9. Yield Components

Yield was evaluated by weighing the ears at a moisture content of 14%. Three components were considered for yield: kilograms of cob per hectare, cob weight, and 1000-seed weight. For the first component (kg∙ha^−1^), all of the cobs harvested from the central rows of each experimental unit were weighed, and projected to kilograms per hectare. Subsequently, 10 cobs were selected at random and weighed individually, obtaining the second yield component (cob weight). Finally, a group of 1000 seeds per experimental unit was randomly selected for the third component (weight of 1000 seeds).

### 2.10. Statistical Analysis

The results were analyzed using R software version 4.3.1 (Lucent Technologies, Murray Hill, NJ, USA) through an analysis of variance (ANOVA) with a significance level of 0.05, after corroborating the assumptions of normality of the obtained data (Shapiro–Wilk test) and homogeneity of variances (Bartlett test). Means were compared using Tukey’s test with a significance level of *p* ≤ 0.05.

## 3. Results

### 3.1. Plant Height and Root Length

The results revealed a significant interaction between inoculation and fertilizer dose on plant height (*p* < 0.001), indicating that both factors influenced plant growth (Figure 2). Additionally, the quadratic effect of nitrogen fertilizer dose was significant for the inoculation factor (*p* = 0.0014). As shown in Figure 2, non-inoculated plants experienced a rapid increase in height with increasing fertilization; however, this growth plateaued at higher doses (>90 kg N∙ha^−1^). In contrast, inoculated plants showed consistent height increases, even at the highest nitrogen dose (120 kg N∙ha^−1^).

The combined treatment of *A. brasilense* inoculation + 120 kg N∙ha^−1^ resulted in the highest plant height, averaging 288 cm, and it was statistically different from the non-inoculated treatment with the same nitrogen dose. Additionally, the inoculated plants + 90 kg N∙ha^−1^ were statistically similar to the anon-inoculated plants + 120 kg N∙ha^−1^. This indicates that, despite a reduction in fertilization, the inoculated plants achieved the same height as the non-inoculated plants (Figure 2). 

No significant interaction was observed for root length; however, each factor had a significant individual effect over the evaluated parameters. The inoculated treatment with *A. brasilense* showed a higher average root length (37.4 cm) than the control, which had an average of 32.07 cm (Figure 3A). Regarding fertilization, root length consistently increased across all evaluated doses, with higher nitrogen levels resulting in longer roots (Figure 3B).

### 3.2. Foliar Nitrogen

The results showed significant interaction between inoculation and nitrogen fertilization (*p*-value = 0.018), with a quadratic effect observed in the interaction. As shown in Figure 4, as the nitrogen fertilization dose increased, the difference between the inoculated and uninoculated treatments diminished. Notably, in the absence of urea fertilization, the inoculated plants with *A. brasilense* outperformed the control and showed values comparable to those of the fertilized treatments. This demonstrates the beneficial role of inoculation, as the rhizobia facilitated biological N fixation, effectively meeting the plants’ nitrogen requirements.

### 3.3. Fresh and Dry Weight

For the fresh and dry biomass parameters, no significant interaction between inoculation and nitrogen fertilization was observed. However, individual analysis of each factor showed a clear effect on the biomass variables. Inoculated plants consistently outperformed the control, with increases of 21.5% in aerial fresh weight, 30% in root fresh weight, 16.9% in aerial dry weight, and 27.7% in root dry weight. Regarding fertilization, the highest biomass values, both fresh and dry, were obtained in plants that received 120 kg N∙ha^−1^ of urea (Table 4).

### 3.4. Cob Proximate Analysis

Cobs proximate analysis results revealed that there was no significant interaction between the factors, nor was there an individual effect of inoculation or nitrogen fertilization on the protein, ash, or moisture content (Table 5). However, the anthocyanin concentration was significantly affected by the fertilization factor. Specifically, treatments with 30 and 90 kg N∙ha^−1^ of urea fertilization showed the highest anthocyanin values, while the control (without urea fertilization) had the lowest concentration of anthocyanins.

### 3.5. Length, Diameter, and Height of Cob

The analysis revealed a significant interaction between factors for the cob length parameters (*p*-value = 0.02) and cob diameter, with a significant quadratic effect observed for the latter (Figure 5). Inoculation with *Azospirillum brasilense* + 120 kg N∙ha^−1^ fertilization resulted in the highest values for both cob length (~16 cm) and diameter (~51 mm). As shown in Figure 5A, the inoculated treatment + 90 kg N∙ha^−1^ fertilization produced similar cob lengths as the non-inoculated + 120 kg N∙ha^−1^ fertilization, indicating that inoculation allowed for a reduction in fertilization doses without compromising cob average length. Similarly, Figure 5B demonstrates that the inoculated treatment with 120 kg N∙ha^−1^ fertilization achieved a greater cob diameter compared to the same dose without inoculation. Furthermore, the inoculation treatment and 90 kg N∙ha^−1^ outperformed the non-inoculated treatments across all evaluated nitrogen levels.

No significant interaction between the two factors was observed for cob height. However, when evaluating the individual effects of each factor, significant differences were found for this parameter. The inoculated treatment with *Azospirillum brasilense* was significantly different from the control (*p*-value = 0.002), achieving an average cob height of 121.88 cm (Figure 6A). Regarding nitrogen fertilization, it was noted that cob height increased at higher fertilization doses. Similar to the results for plant height, at higher nitrogen doses, there was no further noticeable increase in cob height (Figure 6B).

### 3.6. Grains per Cob

No significant interaction was found between the two factors. However, the individual effects of each factor were significant (Figure 7). Inoculation with *Azospirillum brasilense* led to higher grain yields per cob compared to the non-inoculated treatment. For nitrogen fertilization, 120 and 90 kg N∙ha^−1^ doses produced superior and statistically equivalent yields, while the treatment without urea fertilization resulted in the lowest grain yield among all treatments.

### 3.7. Yield 

Significant interaction between the factors was observed for cob weight. The inoculated treatment with *A. brasilense* combined with 120 kg N∙ha^−1^ fertilization distinguished itself from the other treatments, achieving an average cob weight of 168 g, which is 11% higher than the same fertilization dose without inoculation. Additionally, a significant quadratic effect was detected between the factors. As shown in Figure 8, for non-inoculated plants, cob weight increased exponentially at lower fertilization doses (30 and 60 kg N∙ha^−1^); however, at higher doses (90 and 120 kg N∙ha^−1^), the weight plateaued, indicating no further increase with higher fertilization. Conversely, the inoculated plants exhibited a continuous increase in cob weight up to 120 kg N∙ha^−1^, suggesting that further increases in fertilization beyond this dose could still enhance cob weight in the inoculated plants.

For both yield (kg∙ha^−1^) and 1000-seed weight, no significant interaction between the factors was observed, though individual effects were noted. Inoculated plants with *A. brasilense* produced an average of 5241 kg∙ha^−1^ and 523 g/1000 seeds, reflecting increases of 21.8% and 9.4%, respectively, compared to the non-inoculated control (Figure 9A,B). In terms of fertilization, the highest values for both parameters were obtained from plants treated with 120 kg N∙ha^−1^ (Figure 9C,D).

### 3.8. Heatmap Analysis

The heatmap analysis (Figure 10) provides a comprehensive overview of all treatment effects on the evaluated parameters. Two distinct clusters (A and B) are identifiable. Cluster B includes treatments that exhibited the lowest values across most parameters: the non-inoculated control with low fertilization doses (0, 30, and 60 kg N∙ha^−1^), and the inoculated treatment with *A. brasilense* in the absence of urea fertilization. In contrast, Cluster A comprises treatments with the highest values, including all *A. brasilense*-inoculated treatments with fertilization (30, 60, 90, and 120 kg N∙ha^−1^) and non-inoculated treatments with higher fertilization doses (90 and 120 kg N∙ha^−1^).

With stricter grouping, clusters A and B were further subdivided into two distinct clusters each. This analysis differentiates the two most extreme treatments: the non-inoculated control without urea fertilization, which recorded the lowest values across all parameters (Cluster E), and the *A. brasilense*-inoculated treatment with 120 kg N∙ha^−1^ fertilization, which emerged as the best-performing treatment (Cluster C). The control treatment only showed high values when fertilized with more than 90 kg N∙ha^−1^ (Cluster D), as treatments with 30 and 60 kg N∙ha^−1^ fertilization were grouped with the lowest values. In contrast, the *A. brasilense*-inoculated treatments had a positive effect on plants even at the lowest fertilization dose (30 kg N∙ha^−1^). The only inoculated treatment that did not yield positive results was the one without urea fertilization (Cluster F).

## 4. Discussion

*Azospirillum* is one of the most significant genera of diazotrophic bacteria in agriculture [27], with *A. brasilense* being the most extensively studied species. Beyond its nitrogen-fixing capacity, *A. brasilense* is recognized for promoting plant growth through mechanisms such as phytohormone production [28], phosphate solubilization [29], antibiotic production, and induced systemic resistance [11]. In our study, inoculation with *A. brasilense* increased plant height by 10.5% (Figure 2) and root length by 16.7% (Figure 3A). Similar effects have been reported in studies such as Marques et al. [30], which demonstrated a ~44% increase in root length of inoculated maize under both irrigated and water-deficit conditions compared to non-inoculated plants. This enhancement in root growth due to *A. brasilense* has also been observed in other crops like strawberries [31] and wheat [32]. The most pronounced effects of *A. brasilense* inoculation tend to be on root morphology [33]. Larger root systems enable plants to absorb more nutrients, which is reflected in increased plant height. In our study, the combination of *A. brasilense* inoculation with 120 kg N∙ha^−1^ fertilization was statistically superior to the non-inoculated counterparts in terms of plant height. This result contrasts with Oliveira et al. [34], where inoculation combined with mineral fertilization resulted in plants of similar height to those receiving only mineral fertilization. The observed increase in both root length and plant height is likely attributable to 3-indole acetic acid (IAA) production, a key phytohormone produced by *Azospirillum* that regulates cell division, elongation, apical dominance, and tissue differentiation [35]. Further studies are needed to characterize this strain and confirm its potential IAA production to support this hypothesis.

Inoculation with *A. brasilense* outperformed the non-inoculated control in all biomass parameters, including fresh and dry aerial biomass and root biomass (Table 4). This result aligns with previous findings: alterations in root structure, such as increased root length, enhance nutrient uptake, leading to greater overall plant biomass [36]. However, this may not always be the case. For instance, in a study by Calzavara et al. [37] on the inoculation of *Bacillus* and *Azospirillum* in maize, where inoculated plants supplemented with nitrogen exhibited shorter roots compared to non-inoculated plants, the root biomass of the inoculated plants was higher. In contrast, our study found that both root length and root biomass increased with *A. brasilense* inoculation at all fertilization levels.

The most extensively studied characteristic of *A. brasilense* is its ability to fix atmospheric nitrogen in association with plants through a process known as Biological Nitrogen Fixation (BNF). This process directly supplies nitrogen to non-leguminous crops [36]. One direct method to assess the nitrogen transfer and availability to plants is by measuring foliar nitrogen content. In our study, foliar nitrogen concentrations were statistically similar (2.2–2.7 g∙100 g^−1^) across all treatments, except for the non-inoculated control without nitrogen fertilization, which had the lowest value (0.380 g∙100 g^−1^), as shown in Figure 4. Interestingly, inoculation with *A. brasilense* did not have an additive effect on increasing leaf nitrogen content, making it the only parameter where inoculation did not significantly boost foliar nitrogen uptake. However, inoculation appeared to enhance nitrogen absorption in plants that were not treated with urea, as the nitrogen content in inoculated plants without fertilization was statistically equivalent to that of only fertilized plants. Moreover, the difference between inoculated and control treatments diminished at higher nitrogen doses (Figure 4). Galindo et al. [38] similarly noted that agronomic efficiency, calculated from the difference between inoculated and non-inoculated treatments, is higher at lower nitrogen doses, possibly due to soil conditions that favor microbial immobilization of the applied nitrogen. Our foliar nitrogen findings suggest that the differences in vegetative growth were not solely attributable to nitrogen uptake, implying that *A. brasilense* may have employed additional mechanisms to promote plant growth. Further research is required to confirm this hypothesis.

The anthocyanin found in purple maize is a natural pigment from the flavonoid group, responsible for the purple or blue coloration of the leaves, cobs, bracts, and corn grains [39]. This characteristic is influenced by the plant’s genotype, the specific organ where it is produced, and environmental factors such as sunlight, rainfall, and pH [7,40]. As shown in Table 5, the anthocyanin content in the cob and corn grains was not significantly affected by inoculation with *A. brasilense*. Although numerical differences in anthocyanin levels were observed concerning nitrogen fertilization, no consistent pattern emerged, suggesting that anthocyanin content was primarily influenced by the maize variety, independent of inoculation or fertilization. Medina-Hoyos et al. [41] reported that in six purple maize varieties, the average stability value (ASV) of anthocyanin content is largely determined by genotype and environment, with the INIA 615 variety having an ASV of 0.23, making it the most stable. This could explain why anthocyanin levels in this study did not increase as a result of *A. brasilense* inoculation.

Regarding cob parameters, the combined treatment of *A. brasilense* inoculation with 120 kg N∙ha^−1^ resulted in longer cobs compared to the control (Figure 5A). Inoculation treatments increased cob diameter by 6.3% (Figure 5B) and cob height by 8% (Figure 6). These findings align with those of Contreras-Liza et al. [18], who also reported a positive influence of *A. brasilense* inoculation combined with nitrogen fertilization on cob length and diameter in maize. Nonetheless, they only found improvements in those parameters when experimental units were inoculated and treated with 180 kg∙N ha^−1^. Furthermore, reducing the N dose by 50% did not give similar results. On the other hand, our study showed that *A. brasilense* inoculation with 120 kg N∙ha^-1^ achieved the highest values in those variables, surpassing the non-inoculated control. Even when the nitrogen dose was reduced to 90 kg N∙ha^−1^, inoculated plants still outperformed the non-inoculated control with 120 kg N ha^−1^, indicating nitrogen replacement of 30 kg N∙ha^−1^ due to inoculation. Hungria et al. [17] suggest that inoculation with *Azospirillum* in maize can reduce nitrogen requirements by 25%, leading to significant savings. A steady increase in cob parameters was observed across treatments with varying nitrogen levels, reflecting the influence of genetics and abiotic factors.

According to Hungria et al. [17], *A. brasilense* inoculation directly influences the number of grains per cob, as this bacterium not only fixes nitrogen but also participates in additional mechanisms that enhance maize grain yield by approximately 3.1%. This is consistent with our results, as it was observed that inoculation with *A. brasilense* was significantly better than the non-inoculated counterparts in terms of the number of grains per cob (Figure 7A). The average increase was 14.6%.

The combined treatment of *A. brasilense* inoculation with 120 kg N∙ha^−1^ resulted in the highest values for cob weight, significantly differing from all non-inoculated treatments. Additionally, plants inoculated with *A. brasilense* outperformed the control by 21.8% in yield (kg∙ha^−1^) and 9.4% in 1000-seed weight. Similar findings were reported by Hungria et al. [17], who observed a 4.6% increase in grain yield when maize was inoculated with *A. brasilense*; Zeffa et al. [36] demonstrated that *Azospirillum* inoculation improved growth and increased maize yield; and Martinez Reyes et al. [42] showed that the use of *Azospirillum* allowed for a reduction in nitrogen fertilizer use while maintaining the same 1000-seed weight. The combined action of *Azospirillum* strains and chemical fertilization promotes higher 1000-seed weight, as both contribute carbohydrates and macronutrients necessary for grain filling [43].

Several studies have reported a synergy between nitrogen fertilization and *Azospirillum* inoculation [17,34,35]. However, there are cases, such as the study by Marini et al. [44], where no such association was observed. These mixed results show us the need for more research in different locations and with other corn varieties to fully standardize and optimize this sustainable agricultural technology.

The novelty of this study lies in the lack of previous data on *A. brasilense* application in purple maize fertilization (*Zea mays* L.), a highly valued variety not only for its nutritional content but also for its medicinal properties, particularly for its high levels of antioxidants such as anthocyanins. This research offers significant potential benefits by demonstrating that *A. brasilense* inoculation can reduce nitrogen fertilizer use, lowering production costs while enhancing agricultural sustainability and soil health.

Prospects include a broader application of this technology to other regions and maize varieties, thereby expanding its impact on Peruvian agriculture. Long-term studies should focus on inoculation effects on soil quality, microbial biodiversity, and purple maize’s potential to enhance nutrition and human health. Additionally, evaluating the economic and environmental impact of reducing fertilizer use at both regional and national levels will be crucial for fully understanding the benefits of this approach.

## 5. Conclusions

Our results demonstrate that plants inoculated with *A. brasilense* show significant improvements in root length, fresh and dry biomass of both aerial and root parts, cob height, grains per cob, yield (kg∙ha^−1^), and 1000-seed weight. For parameters, namely plant height, leaf nitrogen content, cob weight, cob length, and cob diameter, interaction between inoculation and nitrogen fertilization was observed, with the combined treatment of *A. brasilense* inoculation + 120 kg N∙ha^−1^ fertilization being the most effective. Additionally, a reduction of 30 kg N∙ha^−1^ was possible, as the inoculated treatment with 90 kg N∙ha^−1^ was statistically equivalent to the non-inoculated treatment with 120 kg N∙ha^−1^ in these parameters, except for foliar nitrogen. Reducing dependence on chemical fertilizers helps preserve long-term soil fertility, promotes microbial activity, and balances nutrients in purple maize cultivation.

## Figures and Tables

**Figure 1 microorganisms-12-02107-f001:**
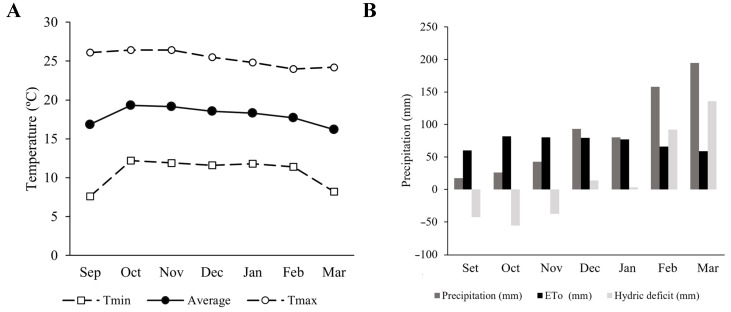
Maximum (Tmax), minimum (Tmin), and average temperature (**A**); precipitation, evapotranspiration (ETo), and hydric deficit per month (**B**) of the experimental site at the National Institute of Agricultural Innovation, Huamanga, Ayacucho, Peru. Data were obtained from the INIA-Canaán Meteorological Station (13°9′ S; 74°13′ W), belonging to the SENAMHI.

**Figure 2 microorganisms-12-02107-f002:**
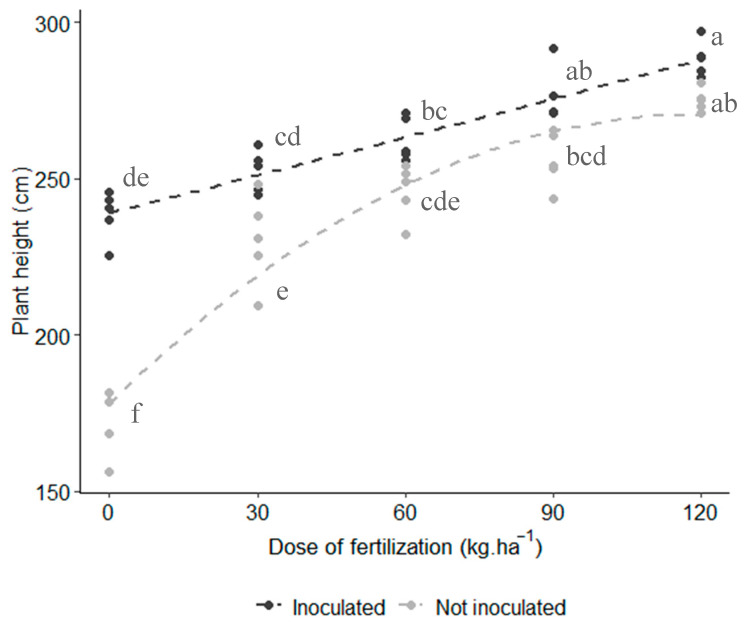
Effect of inoculation and N fertilization on purple corn’s plant height. Interaction plot of two factors with quadratic effect (*p*-value = 0.001). Lines represent the regression trend for each inoculation treatment. Model’s Adjusted R^2^: 0.888. Means with different letters are statistically different (Tukey test, α = 0.05).

**Figure 3 microorganisms-12-02107-f003:**
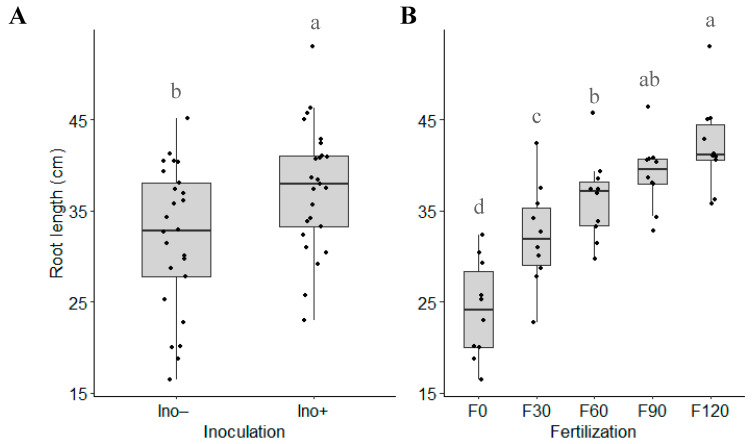
Root length based on (**A**) inoculation and (**B**) N fertilization. (Ino− = not inoculated; Ino+ = inoculation with *A. brasilense*; F0 = without urea fertilization; F30 = 30 kg N∙ha^−1^; F60 = 60 kg N∙ha^−1^; F90 = 90 kg N∙ha^−1^; F120 = 120 kg N∙ha^−1^). Means with different letters in the same factor are statistically different (Tukey’s test, α = 0.05).

**Figure 4 microorganisms-12-02107-f004:**
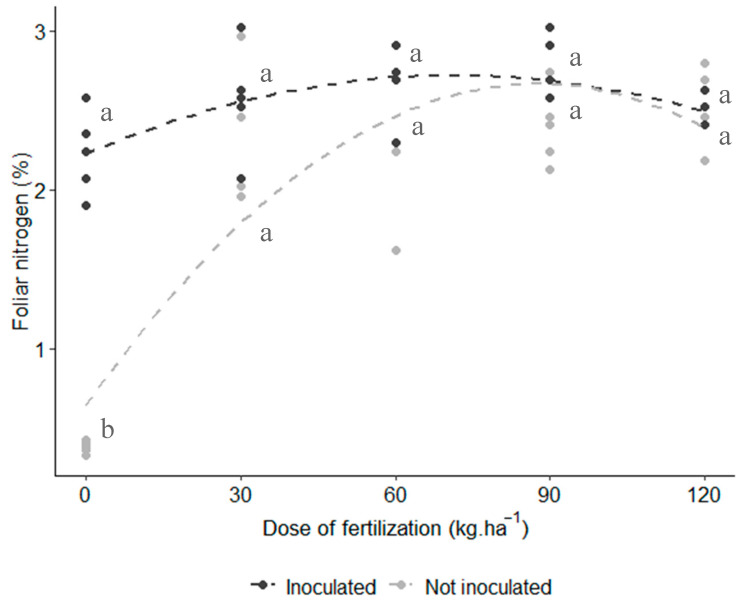
Effect of inoculation and urea fertilization on purple corn’s foliar nitrogen. Interaction plot of two factors with quadratic effect (*p*-value = 0.018). Lines represent the regression trend for each inoculation treatment. Model’s adjusted R^2^: 0.726. Means with different letters are statistically different (Tukey’s test, α = 0.05).

**Figure 5 microorganisms-12-02107-f005:**
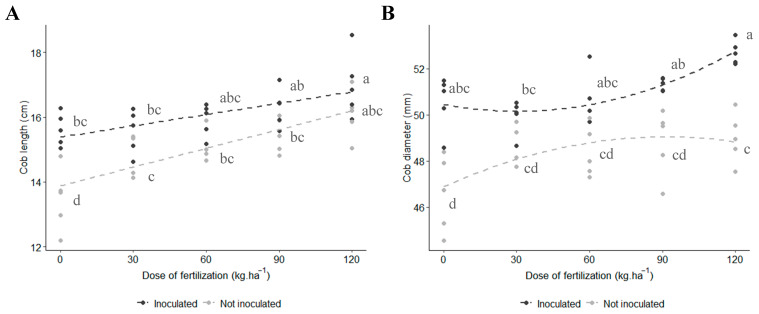
Effect of inoculation and N fertilization on the cob length and diameter of purple corn. (**A**) Interaction plot of two factors with linear effect (*p*-value = 0.022). Lines represent the regression trend for each inoculation treatment. Model’s adjusted R^2^: 0.593. (**B**) Interaction plot of two factors with quadratic effect (*p*-value = 0.006). Lines represent the regression trend for each inoculation treatment. Model’s adjusted R^2^: 0.684. Means with different letters are statistically different (Tukey’s test, α = 0.05).

**Figure 6 microorganisms-12-02107-f006:**
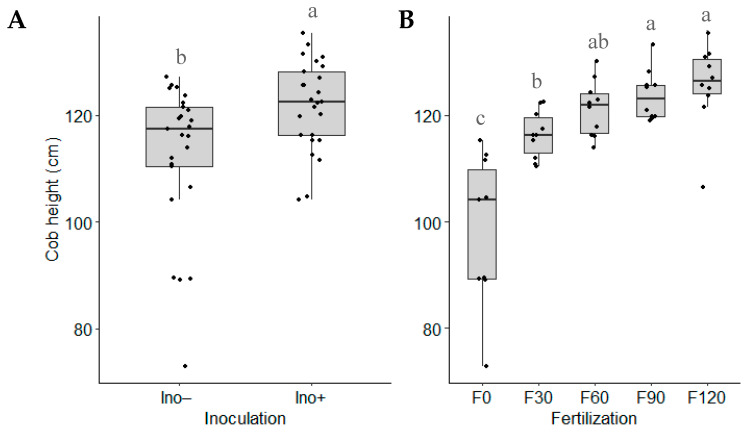
Cob height based on (**A**) inoculation and (**B**) N fertilization (Ino− = not inoculated; Ino+ = *A. brasilense* inoculation; F0 = without urea fertilization; F30 = 30 kg N∙ha^−1^; F60 = 60 kg N∙ha^−1^; F90 = 90 kg N∙ha^−1^; F120 = 120 kg N∙ha^−1^). Means with different letters in the same factor are statistically different (Tukey’s test, α = 0.05).

**Figure 7 microorganisms-12-02107-f007:**
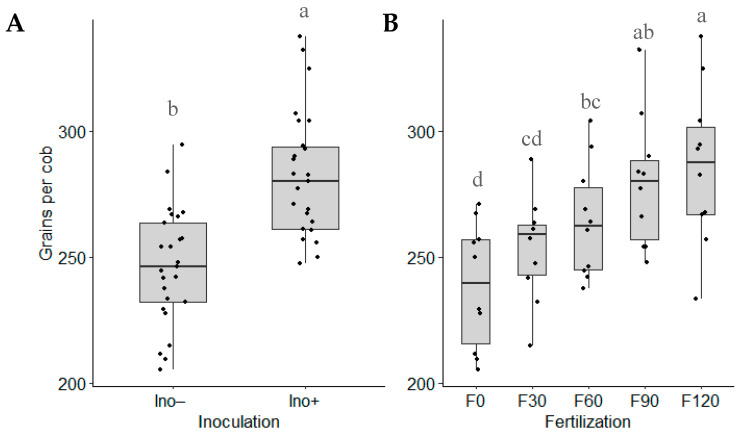
Grains per cob based on (**A**) inoculation and (**B**) N fertilization (Ino− = not inoculated; Ino+ = *A. brasilense* inoculation; F0 = without urea fertilization; F30 = 30 kg N∙ha^−1^; F60 = 60 kg N∙ha^−1^; F90 = 90 kg N∙ha^−1^; F120 = 120 kg N∙ha^−1^). Means with different letters in the same factor are statistically different (Tukey’s test, α = 0.05).

**Figure 8 microorganisms-12-02107-f008:**
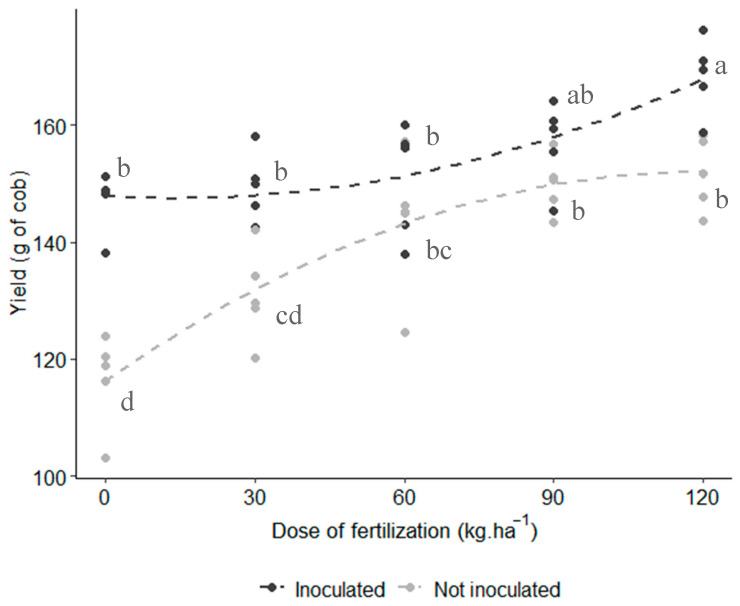
Effect of inoculation and N fertilization on the yield (cob weight) of purple corn. Interaction plot of two factors with quadratic effect (*p*-value = 0.003). Lines represent the regression trend for each inoculation treatment. Model’s adjusted R^2^: 0.770. Means with different letters are statistically different (Tukey’s test, α = 0.05).

**Figure 9 microorganisms-12-02107-f009:**
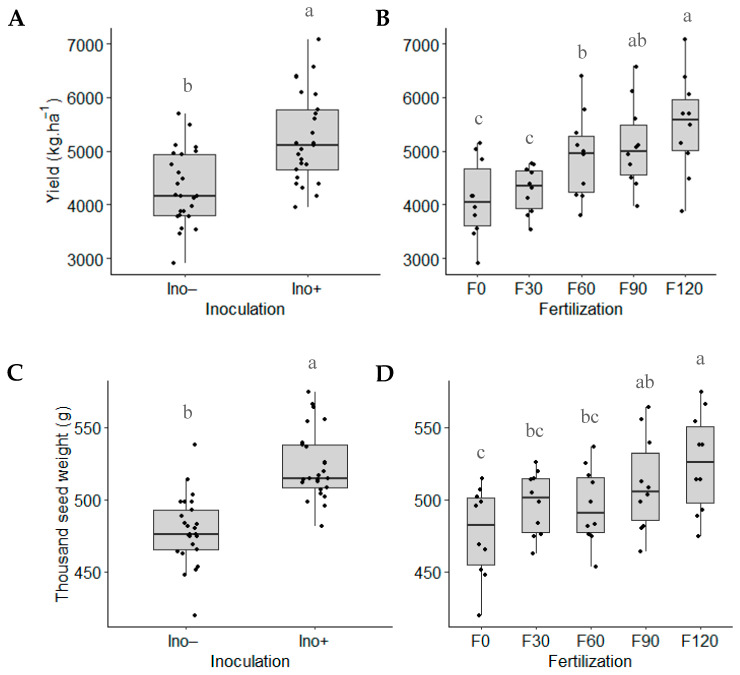
Yield (kg∙ha^−1^) and 1000-seed weight of the purple corn based on (**A**,**C**) inoculation and (**B**,**D**) N fertilization (Ino− = not inoculated; Ino+ = *A. brasilense* inoculation; F0 = without urea fertilization; F30 = 30 kg N∙ha^−1^; F60 = 60 kg N∙ha^−1^; F90 = 90 kg N∙ha^−1^; F120 = 120 kg N∙ha^−1^). Means with different letters in the same factor are statistically different (Tukey’s test, α = 0.05).

**Figure 10 microorganisms-12-02107-f010:**
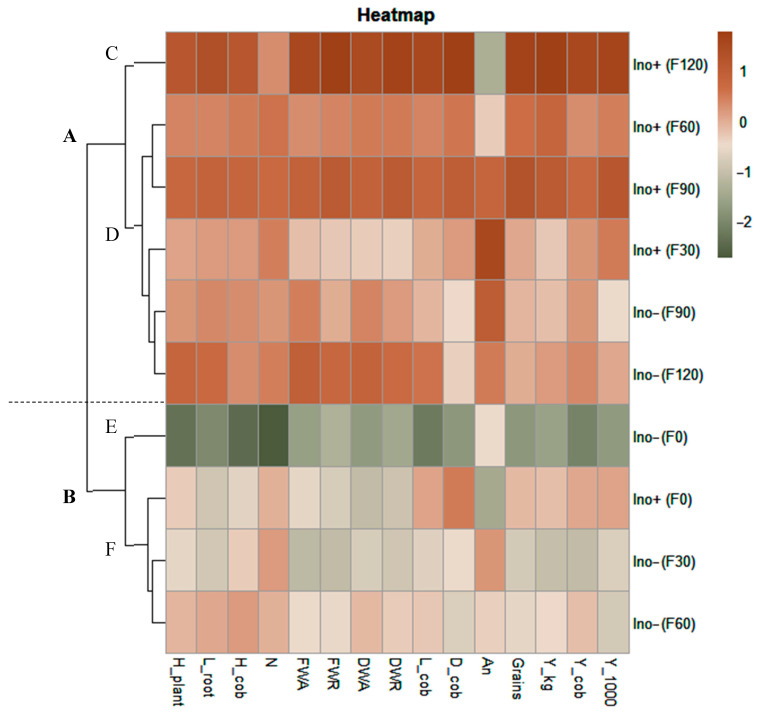
Heatmap and cluster analysis of the combined effects between inoculation and N fertilization. The column corresponds to the treatments and the line to the evaluated parameters. Clusters are represented by the letter A–F. (H_plant = plant height; L_root = root length; H_cob = cob height; N = foliar nitrogen; FWA = fresh weight aerial; FWR = fresh weight root; DWA = dry weight aerial; DWR = dry weight root; L_cob = cob length; D_cob = cob diameter; An = anthocyanins; Grains = grains per cob; Y_kg = yield expressed in kg per ha; Y_cob = yield expressed in cob weight; Y_1000 = yield expressed in weight of a thousand seeds).

**Table 1 microorganisms-12-02107-t001:** Physicochemical characterization of the soil in the experimental plot.

Variable	Units	Result
Sand	%	48
Silt	%	40
Clay	%	12
Texture	-	Sandy loam soil
pH	-	7.9
Electrical conductivity	mS∙m^−1^	10.1
Organic matter	%	2.1
N	%	0.11
P	Ppm	27.84
K	Ppm	359.86
Ca	Cmol(+)∙kg^−1^	18.3
Mg	Cmol(+)∙kg^−1^	2.52
K	Cmol(+)∙kg^−1^	0.37
Na	Cmol(+)∙kg^−1^	0.12
CEC	Cmol(+)∙kg^−1^	21.3

**Table 2 microorganisms-12-02107-t002:** Treatment combination. Levels of factor 1 and 2 show an abbreviation code between brackets.

Factor 1. Inoculation	Factor 2. N Fertilization (kg N∙ha^−1^)
Not inoculated (Ino−)	0 (F0)
	30 (F30)
	60 (F60)
	90 (F90)
	120 (F120)
Inoculated with *Azospirillum brasilense* (Ino+)	0 (F0)
	30 (F30)
	60 (F60)
	90 (F90)
	120 (F120)

**Table 3 microorganisms-12-02107-t003:** Total N (kg∙ha^−1^) for each level of factor 2.

Level	Urea	N from Urea *	N from Base Fertilization	Total N
	kg∙ha^−1^	kg∙ha^−1^	kg∙ha^−1^	kg∙ha^−1^
F0	0	0	186.6	186.6
F30	65.22	30	186.6	216.6
F60	130.43	60	186.6	246.6
F90	195.65	90	186.6	276.6
F120	260.86	120	186.6	306.6

* Urea total dose was split in 2 applications (1st: at sowing; 2nd: at 47 das).

**Table 4 microorganisms-12-02107-t004:** Fresh and dry weight of the aerial and root biomass based on inoculation and N fertilization.

Treatment	Fresh Weight		Dry Weight	
	Aerial	Root	Aerial	Root
Interaction	n.s.	n.s.	n.s.	n.s.
Factor 1. Inoculation				
Ino−	613 ± 181 b	63 ± 19 b	142 ± 41 b	18 ± 6 b
Ino+	745 ± 143 a	82 ± 22 a	166 ± 41 a	23 ± 7 a
Factor 2. N fertilization				
F0	492 ± 125 e	50 ± 9 d	96 ± 19 e	12 ± 3 e
F30	563 ± 113 d	58 ± 13 d	131 ± 13 d	16 ± 3 d
F60	660 ± 99 c	71 ± 14 c	160 ± 19 c	21 ± 4 c
F90	795 ± 53 b	84 ± 17 b	180 ± 14 b	25 ± 5 b
F120	887 ± 72 a	101 ± 15 a	201 ± 19 a	29 ± 5 a

Means with different letters in the same factor for each parameter are statistically different (Tukey’s test, α = 0.05). n.s. no significance. (Ino− = not inoculated; Ino+ = *A. brasilense* inoculation; F0 = without urea fertilization; F30 = 30 kg N∙ha^−1^; F60 = 60 kg N∙ha^−1^; F90 = 90 kg N∙ha^−1^; F120 = 120 kg N∙ha^−1^).

**Table 5 microorganisms-12-02107-t005:** Proximate analysis parameters in purple maize cobs based on inoculation and N fertilization.

Treatment	Anthocyanins	Proteins	Ash	Moisture Content
	g∙100 g^−1^	g∙100 g^−1^ N × 6.25	g∙100 g^−1^ b.s.	g∙100 g^−1^
Interaction	n.s.	n.s.	n.s.	n.s.
Factor 1. Inoculation				
Ino–	0.092 ± 0.03 n.s.	8.91 ± 0.95 n.s.	1.91 ± 0.12 n.s.	11.64 ± 0.68 n.s.
Ino+	0.086 ± 0.04	8.94 ± 0.66	1.98 ± 0.11	11.74 ± 0.45
Factor 2. N fertilization				
F0	0.071 ± 0.02 b	8.82 ± 0.84 n.s.	1.92 ± 0.10 n.s.	11.41 ± 0.49 n.s.
F30	0.106 ± 0.03 a	8.99 ± 0.81	1.94 ± 0.12	11.50 ± 0.37
F60	0.083 ± 0.02 ab	8.58 ± 0.58	1.93 ± 0.13	11.74 ± 0.51
F90	0.107 ± 0.04 a	9.13 ± 1.06	1.95 ± 0.15	11.75 ± 0.53
F120	0.081 ± 0.03 ab	9.12 ± 0.69	1.99 ± 0.11	12.03 ± 0.79

Means with different letters in the same factor for each parameter are statistically different (Tukey, α = 0.05). n.s. no significance. (Ino− = not inoculated; Ino+ = *A. brasilense* inoculation; F0 = without urea fertilization; F30 = 30 kg N∙ha^−1^; F60 = 60 kg N∙ha^−1^; F90 = 90 kg N∙ha^−1^; F120 = 120 kg N∙ha^−1^).

## Data Availability

The data presented in this study are available from the corresponding author upon request.

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
