# Peer review of "Inoculation with Azospirillum brasilense as a Strategy to Reduce Nitrogen Fertilization in Cultivating Purple Maize (Zea mays L.) in the Inter-Andean Valleys of Peru"

_microorganisms, 2024, doi:10.3390/microorganisms12102107_

Round 1
Reviewer 1 Report
Comments and Suggestions for Authors
(1) Purple maize has gained global significance due to its numerous nutraceutical benefits. However, sustaining its production typically requires high doses of nitrogen fertilizers, which, when applied in excess, can contaminate vital resources such as soil and water. Inoculation with nitrogen-fixing microorganisms, such as those from the Azospirillum genus, has emerged as an alternative to partially or fully replace nitrogen fertilizers. The study aimed to evaluate the inoculation effect with A. brasilense and varying nitrogen fertilization levels on the yield and quality of purple maize. The experiment was carried out using a randomized complete block design (RCBD) with a 2 x 5 factorial arrangement and 5 replications. Treatments comprised two inoculation levels under five nitrogen doses. Inoculation with A. brasilense resulted in a 10.5% increase in plant height, a 16.7% increase in root length, a 21.3% increase in aboveground fresh biomass, a 30.1% increase in root fresh biomass, and a 27.7% increase in leaf nitrogen concentration compared to the non-inoculated control. After carefully reading:
(2) The theme of this paper is of good interest and worthy of investigation. The structure and organization are also good. However, there are still some issues required to be addressed to improve the manuscript.
(3) Fig. 1C: What does the word “ETo” mean? Please explain it at the end of the figure title.
(4) Section 2.2: The soil components of soil sample should be clarified to determine the soil type, e.g.., contents of clay and silt.
(5) Table 2: The abbreviations of “Without inoculation” and “Inoculation with Azospirillum brasilense” should be revised to avoid the confusion of letters A and C in Table 2 and these in figure title.
(6) Section 2.9: The specific measuring method of cobs yield should be explained in detail. Additionally, were 10 cobs selected at only one location in measuring cob length, diameter, weight and 1000-seed weight?
Reviewer 2 Report
Comments and Suggestions for Authors
This paper investigated the combined effects of inoculation and nitrogen fertilization on the growth and quality parameters of purple maize, the topic is interesting and the data is solid. The experimental methods and results are well-addressed. However, the Introduction needs to be extended by reviewing the literature regarding inoculation research in maize applied with various nitrogen doses. Below are some comments that help improve this paper:
Figure 1C: add the x-axis.
The Material and Methods were clearly and exhaustively addressed.
Figure 2: Correct the formation of unit kg.ha-1.
I suggested changing Figures 2, 5, and 8 into bar graphs since the line doesn’t make sense when comparing the differences between treatments.
Lines 597-600: these sentences are repeated from the results.
The conclusion needs to be shortened. Paragraphs 2 and 3 can be moved to the discussion section.
Comments on the Quality of English LanguageNo issues were detected.
